# Post-Procedural Follow-Up of the Interventional Radiology’s Management of Osteoid Osteomas and Osteoblastomas

**DOI:** 10.3390/jcm11071987

**Published:** 2022-04-02

**Authors:** Chiara Acanfora, Enrico Grassi, Giuliana Giacobbe, Marilina Ferrante, Vincenza Granata, Antonio Barile, Salvatore Cappabianca

**Affiliations:** 1Diagnostic and Interventional Radiology, Department of Biotechnology and Applied Clinical Science, University of L’Aquila, 67100 L’Aquila, Italy; antonio.barile@univaq.it; 2Division of Radiology, Department of Precision Medicine, University of Campania Luigi Vanvitelli, 80138 Naples, Italy; e.grassipercesepe@gmail.com (E.G.); gi.giacobbe10@gmail.com (G.G.); marilina.ferrante@studenti.unicampania.it (M.F.); salvatore.cappabianca@unicampania.it (S.C.); 3Radiology Division, Istituto Nazionale Tumori IRCCS Fondazione Pascale-IRCCS di Napoli, 80131 Naples, Italy; v.granata@istitutotumori.na.it

**Keywords:** osteocytic tumor, interventional radiology, MRgFUS, radiofrequency, ablation, osteoid osteoma, osteoblastoma, follow-up

## Abstract

The family of painful osteocytic tumors includes osteoblastomas and osteoid osteomas—these lesions are considered benign, but they could produce a significant painful symptomatology. Usually, people affected are between 20 s and 30 s. When symptomatic, an effective treatment is mandatory for the management of these lesions to allow for a ful quality of life. The possibilities of treatment range from chirurgical en-block resection (procedure of surgical oncology aiming to remove a tumoral mass in its entirety, completely surrounded by a continuous layer of healthy tissue) to interventional approaches that, nowadays, are considered the most affordable and sustainable in terms of effectiveness, recovery after procedure, and for bone structure sparing. The main techniques used for osteoid osteomas and osteoblastomas are radio frequency ablation (RFA) and magnetic resonance-guided focused ultrasound (MRgFUS): the most important difference between these approaches is the needleless approach of MRgFUS, which further reduces the minimal invasiveness of RFA (and the related consequences) and the absence of exposure to ionizing radiation. Despite their high efficacy, a recurrence of pathology may occur due to a failure in therapy. In light of this, describing the various possibilities of follow up protocols and the imaging aspects of recurrence or incomplete treatment is mandatory. In the scenario given in the literature, many authors have tried to asses an organized follow up protocol of these patients, but many of them did not undergo periodical magnetic resonance (MR) or computerized tomography (CT) because of the lack of symptomatology. However, even if it seems that clinical evolution is central, different papers describe the protocol useful to detect eventual relapse. The aim of our manuscript is to review the various possibilities of follow-up of these patients and to bring together the most salient aspects found during the management of these osteocytic bone lesions.

## 1. Introduction

Among the family of benign osteocytic tumors, we can include osteoblastomas, osteoid osteomas, osteomas, and bone islands. These lesions are benign, osteogenic bone tumors that appear primarily in young patients in their 20s and 30s. A bone island or enostosis is a focus of mature compact (cortical) bone within the cancellous bone (spongiosa), reflecting a developmental error during the process of endochondral ossification. The lesion is typically asymptomatic, manifests itself with highly distinctive radiologic features, and it does not need invasive or mini-invasive treatment because of its lack of symptoms. Similarly, osteoma is a benign osteogenic lesion characterized by the proliferation of compact or cancellous bone. It can be central, peripheral, or extraskeletal. When symptomatic, treatment consists of complete surgical removal at the base where it unites with the cortical bone. The two lesions of benign osteocytic tumor groups that today are routinely treated by the interventional radiology (IR) approach are osteoid osteomas and osteoblastomas. The most important difference between these lesions is primarily their dimensions: the osteoid osteoma’s diameter is less than 1.5 cm, and the osteoblastoma exceeds this size [1]. Osteoid osteomas are relatively frequent benign bone tumors (10 to 20% of benign bone tumors) that most often occur during the first 30 years of life (90% of patients are less than 25 years old), with male predominance (male/female ratio of 2:1). Osteoblastoma is an uncommon neoplasm that represents 3% of all benign bone tumors and about 1% of all primary bone neoplasms. Osteoblastoma usually occurs in young patients (10–25 years), with a male/female ratio of 2:1.

## 2. Osteoid Osteoma

Osteoma osteoid is characterized by the presence of a nidus that is composed of immature bone, and is full of blood vessels and nerve cells that deliver prostaglandins, especially prostaglandin E2, which is possibly considered the cause of the peculiar night-time pains [2]. Usual radiological signs of osteoid osteoma are the central, radiotransparent, partly ossifying nidus measuring from a minimum of 2–3 mm to a maximum of 15 mm, and encircled by a zone of reactive ossification [3]. If not treated, osteoid osteomas show both clinical and radiological natural decline. Nevertheless, this course takes usually 6 years (range 2 to 15 years). NSAIDS not only mitigate the manifestations in patients with osteoid osteoma, but also speed up the clinical setback of the symptoms [4,5,6]. Treatment is frequently attempted early, because pain can affect daily activities and because the medical approach constitutes only symptomatic management and can cause long-lasting side effects. On the light of this, we can consider different invasive and non-invasive approaches that are basically directed to the aggression of the nidus. Regarding an invasive approach, historically, surgical intervention consisted of en-block resection or open curettage of the nidus, but is currently no longer used [7], on the basis of difficulty in intraoperative visualization of the lesion, which brings significant bone resection and harm to the adjacent tissue; morbidity is connected to weakness of the residual bone and protracted recovery times [8]. In this scenario, mini invasive approaches (especially, percutaneous computed tomography (CT)-guided radio frequency ablation (RFA) [9]) and magnetic resonance-guided focused ultrasound ablation (MRgFUS) [10,11] have become the gold standard techniques for management of these lesions. Radio frequency ablation (RFA) is a good therapeutical tool to replace surgery and to become a valid standard of treatment in light of the fact that it is less invasive, and it has less collateral damage and morbidity [12,13]. During RFA, a needle is placed into the osteoid osteoma lesion with the use of computed tomography (CT) imaging, and is brought to 90 °C to ablate the nidus—thermal ablation eliminates pain instantly [14]. Although RFA has an 80–98% success rate [15], treatment is minimally invasive and can possibly cause collateral tissue harm alongside exposing patients and operators to ionizing radiation [16,17]. On the light of this, the use of MRgFUS is more convenient for avoiding the minimal invasiveness of RFA (and the related consequences) and the exposure of patients and physicians to ionizing radiation. Furthermore, MRgFUS gives the opportunity to supervise the temperature during the whole procedure in order to preserve important and critical structures nearby the lesion: low-power test sonications are performed to check MRgFUS targeting precision to evaluate the power needed for therapeutic sonications. Clinical response is the primary outcome to assess the effectiveness of the IR treatment. Furthermore, finding a valid method to declare the good outcome of these procedures can be useful because of many reasons:To confirm by imaging the clinical suspect of recurrence or incomplete treatment.To demonstrate, eventually, any kind of complications.To look for a possible differential diagnosis in case of a partial or incomplete clinical response.

Many have authors tried to lead studies about the best possibilities for follow-up for these patients, and this study’s aim is to bring together the most salient aspects found during the management of osteocytic bone lesions.

Gonca E. et al. [18] evaluated the imaging modifications between follow-up periods. The aims of this study were to record the long-term outcomes of osteoid osteoma patients and to establish the dynamic contrast-enhanced MR and CT imaging features of RFA treatment-related modifications of osteoid osteomas (OO) between follow-up intervals. In their study, none of the participants presented symptoms that could be connected to OO after the RFA procedure during the long-term follow-up, and their number of successful cases was 100%. Their study population was made up of 30 patients who had the CT-guided RFA approach; the RFA patients were called for follow-up imaging with CT and MRI 1, 3, 6, and 12 months after treatment. In their population of study, there was a considerable disparity for OO nidi sizes between the pretreatment and second follow-up period examinations, there was no statistically significant difference for cortical thickening between pre and post-treatment examinations, and contrast-enhanced-MR imaging after RFA treatment for OO demonstrated that OO nidi continued to be enhanced on post-treatment MRI. Nevertheless, the signal intensity showed a progressive decrease following the months that passed after the procedure during the follow-up. So, although none of the patients of the Gonca et al. study had recurrence or unsuccessful treatment procedures, they were able to presume that patients whose osteoid osteoma nidus did not demonstrate a vanishing in signal intensity upon post-treatment MRI may be a suitable for reablation. In the end, according to the clinical observation, Gonca et al. assumed that a single dynamic contrast-enhanced MRI during the first 3 months after treatment may be adequate for symptom-free patients. Similarly, the Rehnitz C. et al. [19] study followed three purposes: the assessment of the clinical outcome of CT-guided RFA in OO, the evaluation of the morphologic CT and MRI changes of OO pre- and postRFA, and the correlation between imaging changes with markers of clinical success. Their follow-up examinations were typically performed three months after RFA. The long-term outcome was in qualitative terms and quantitative terms, and was evaluated using a standardized survey with a time interval of at least three months after RFA. All the patients of their study underwent CT imaging before and immediately after the procedure, evaluating the same parameters before and after the procedure: location, size, and volume of the lesion; presence or absence of calcification; and cortical thickening. A routine MRI follow-up was recommended, but despite this advice, some patients did not receive an MRI follow-up. In conclusion, however, Rehnitz C. et al. were not able to find specific MR or CT findings that could be correlated with clinical outcome. They suggested performing MRI in the case the symptoms of relapse were present so as to detect nidus changes. Similar to Rehnitz C. et al., B. Gebauer et al. [20] considered analgesia as the most important characteristic in patients who underwent RFA during their long-term follow-up; in fact, a clinical follow-up testing was performed on all of the patients and, even if imaging for the aim of assessing the success of ablation was recommended, it was not regularly performed, since the clinical symptoms of the patients were cardinal. In light of this, contrast-enhanced MRI was performed on various patients, as with any patient in which relapse or other complications were supposed, looking for the nidus enhancement. Furthermore, to determine the long-lasting clinical success of the procedure, a normalized telephone examination for all of the treated patients was performed. This follow-up pathway led them to establish that, in consideration of therapy satisfaction and the personal sense of pain, all of their examined patients were totally satisfied with RFA for the treatment of their osteoid osteoma, without unnecessary imaging follow-up. F. Arrigoni et al. [21] evaluated the evolution of the imaging features of osteoid osteoma treated with RFA or MRgFUS during a long-term follow-up. Their purpose was not to prove the effectiveness of the procedures, so they included only patients successfully treated in their unit, regardless of the kind of treatment executed—both RFA and MRgFUS. The clinical results are the first data that connect with the result of the procedure, but, as was previously shown, considering the other authors’ points of view as well, imaging aspects of the post-treatment evolvement of these treatments were not yet methodically encoded. So, their intent was to analyze and recapitulate a few of characteristic imaging features that identified the imaging at the moment of the diagnosis and that evolution along the follow-up after treatment. They have detected and recorded four easy-to-find signs that demonstrated the effectiveness of the procedure, which they schematically summarized as follows: disappearance of bone marrow oedema around the lesion, regression of perilesional flogistic/synovial reaction, and restructuring of the bone and the “ring sign”. The first two, particularly, indicate the lack of biological activeness under the approached lesion. In line with their experience, at 6 months, even if there was substantially an impressive decrease of these flogistic signs, with their vanishing in up to 53% of cases, it was still feasible to search for these signs in the follow-up MRI. Regarding the kind of instrumental procedure utilized, they proved that MRI is advantageous at the outset, because of its capability of detecting all the reactive features (bone marrow oedema, synovitis, and perilesional reaction), while the employment of the CT scan can be postponed to 1 year, escaping the risk of exposing young patients to a disproportionate amount of radiation. The CT scan is helpful to detect bone remodeling for the bone restoration of the skeletal compartment, which can be found in the long-lasting follow-up. Assessment of the follow-up using these aspects has a pragmatic benefit in clinical practice, as in cases of doubt of disease relapse, for patients who are satisfactorily treated and are thus free from grief, they generally have a lack of observance for undergoing to MRI and CT scans for follow-up.

## 3. Osteoblastoma

Osteoblastoma is an uncommon neoplasm that represents the 3% of all benign bone tumors and about 1% of all primary bone neoplasms. Osteoblastoma usually occurs in young patients (10–25 years), with a male/female ratio of 2:1; these characteristics are similar to osteoid osteoma’s ones [22]. In contrast to osteoid osteoma, nevertheless, osteoblastoma is most frequently localized in the posterior elements of the vertebral column and the sacrum. In addition, we can find osteoblastomas in long bones, craniofacial bones, hands and feet, ribs, clavicle, and sternum [22,23]. Clinically, osteoblastoma does not demonstrate the usual introductory symptoms observed in patients with osteoid osteoma. In patients with osteoblastoma, the most usual presentation is pain, which is typically described as tedious, grieving, and often increasing in severity. Usually, the pain does not react to NSAIDs and is not generally more acute during the night. The radiographic features of osteoblastoma depend on the localization and maturation of the tumor: usually, these tumors appear as radiolucent lesions characterized by an irregular configuration and encircled by a thin framework of reactive bone. The inner of the neoplasm may demonstrate several stages of ossification, depending on the lesion’s maturation. In comparison with osteoid osteoma, osteoblastoma is bigger, with less reactive bone sclerosis and more cortical enlargement. CT examination is the imaging method to choose to characterize osteoblastoma. Typically, the lesion shows zones of mineralization centrally, bone reshaping, and features of reactive sclerosis on the boundary with a thin marginal bone shell. MRI characteristics are non-characteristics and unclear, and there may be an overevaluation of the degree and kind of tumor as a consequence of the augmented regional inflammatory reaction and wide marrow edema. However, MRI is the best tool when the tumor has extensive effects on the spinal canal and cord. The lesion in general appears as a low or intermediate signal intensity on T1-weighted images and intermediate to high signal intensity on T2-weighted images. Osteoblastoma can be approached through non-invasive techniques such as RFA and MRgFUS, but lesions that show locally destructive and aggressive behavior must be managed surgically because of their growth potential. In many patients, intralesional curettage is executed with success; nevertheless, the relapse rate is quite high. En-bloc resection has to be preferred for a conclusive treatment of wide and bone-destructing osteoblastomas, as well as for relapses of diseases after curettage, when possible. Today, however, the most usual treatment for the majority of OB is the mini-invasive technique based on RFA or/and MRgFUS (performed as described above for OO), which can be considered harmless and risk-free, as already demonstrated in the literature [24]. Similar to osteoid osteomas, the clinical response is the primary outcome to assess the effectiveness of the IR treatment. Furthermore, finding a valid method to declare the good outcome of these procedures can be useful because of the reasons previously listed.

F. Arrigoni et al., in 2018 [25], evaluated the effectiveness of the RFA approach in OB treatment predominantly affecting the spine: the purposes of this article were to assess the efficiency and security of the management of spinal OB with CT-guided RFA and to assess the long-lasting outcome considering imaging characteristics. An IR approach should be examined in light of the fact that surgery needs longer hospitalization times and does not always assure clinical effectiveness after the first operation and, moreover, occasionally, supplementary surgical approaches are necessary to obtain stabilization of the spine [26]. In their study, brilliant impacts were obtained in terms of outcome. The use of temporary corticosteroid treatment instantly after the procedure confined the outbreak of inflammatory reactions induced by thermal ablation. In the MRI follow-up studies performed at 6 and 12 months, the lack of reactive inflammatory processes nearby the lesions (also meant as bone edema) and the absence of contrast enhancement was observed. These characteristics validated the success of the procedure. CT images performed at 12 months showed ossification of the lesion in all patients. Checking the progression of the target area over time, gradual bone reshaping was observed. These findings validated that RF thermal ablation does not impact the normal evolution of the spine. Rehnitz et al., in 2012 [24], aimed to retrospectively assess the clinical long-term success of CT-guided RFA in patients with OO and OB. In a retrospective cohort study, they included 77 patients who underwent CT-guided RFA procedures for OO or OB between January 2003 and February 2011. A nidus size exceeding 15 mm was classified as OB. Follow-up examinations were performed 3–6 months later. Outcome was assessed using a standardized questionnaire. A local recurrence of the OO was clinically confirmed by CT and MRI. Thereby, CT-guided RFA was again performed using a standard technique. If after 3 months the pain was only modestly decreased and follow-up MRI validated perseverance of a contrast-enhancing nidus, the RFA was, for this reason, considered as not complete. Anyway, as a retrospective study, even if Rehnitz et al. demonstrated that RFA is a reliable, efficient and long-lasting approach of patients with OO and OB, they did not give a wider and more complete view about a structured and timed follow-up of these patients. Similarly, F. Arrigoni et al. [11] evaluated the treatment of non-spinal intra-articular osteoblastoma by MRgFUS in terms of feasibility, safety, and outcomes. In their study, although MRI and CT scans were evocative of OB (with signs of intense inflammation), for the purpose of acquire histological diagnoses, all patients underwent CT-guided biopsy, which validated the diagnosis. From a clinical point of view, very important results were achieved in the current series with respect to pain relief. Because the lesion itself is to blame for pain, vanishing of bone pain is the first biological sign of the accomplishment of the procedure. In the imaging analysis, they marked an important decrease in bone edema and synovitis, as demonstrated in the previous study [25] with the total settlement. These aspects validate the efficacy of the procedure and, particularly, the complete ablation of the lesion.

## 4. Conclusions

The review of the literature shows certain variability in the follow-up patients. Many groups of authors have followed their patients based on pain assessment, with others using imaging follow-up based on MR and CT (Figure 1 and Figure 2). Because pain evaluation is a non-objective approach, imaging may seem indispensable to reject the possibility of relapse in patients with suspicious/atypical pain. However, treatment-related modifications may be misleading in terms of imaging. So far, it is crucial to decide under what heading radiologists have to decide if patients can be considered healed or not after the treatment. Finally, even though the follow-up of the barely invasive approach of osteoid osteoma is primarily clinical, the follow-up with MRI and CT scans can be helpful in the face of perseverance of grief or clinical suspect of relapse. MRI within the first 12 months to evaluate the vanishing and/or the fading of the flogistic reaction that constantly is present with the lesion should be considered as the best protocol to verify the effectiveness of the procedure; after 12 months, a CT scan of the skeletal compartment could be helpful to evaluate the bone reshaping and the bone restoration with the vanishing of the approached lesion. For what concerns osteoblastomas, the authors performed MRI. Follow-up at 6 and 12 months and a lack of reactive inflammatory processes nearby the lesions and contrast enhancement was observed. This evidence verified the brilliant outcome of the procedure. CT images performed at 12 months demonstrated ossification of the lesion in all patients. Observing the changes of the target area over time, gradual bone reshaping was recognized. These findings validate that RF thermal ablation does not impact the normal evolution of the spine. Anyway, because the lesion itself is to blame for pain, vanishing in bone pain is the first biological sign of the accomplishment of the procedure. In the end, clinical symptomatology seems to make DE rule, but with the possibility of pain recurrence and so of pathology recurrence, only imaging (MR and CT) can ensure the presence or not of the nidus and the surrounding flogistric reaction. Furthermore, MRI and CT can be very helpful to evaluate the bone reshaping that may come from RFA or MRgFUS treatment.

## Figures and Tables

**Figure 1 jcm-11-01987-f001:**
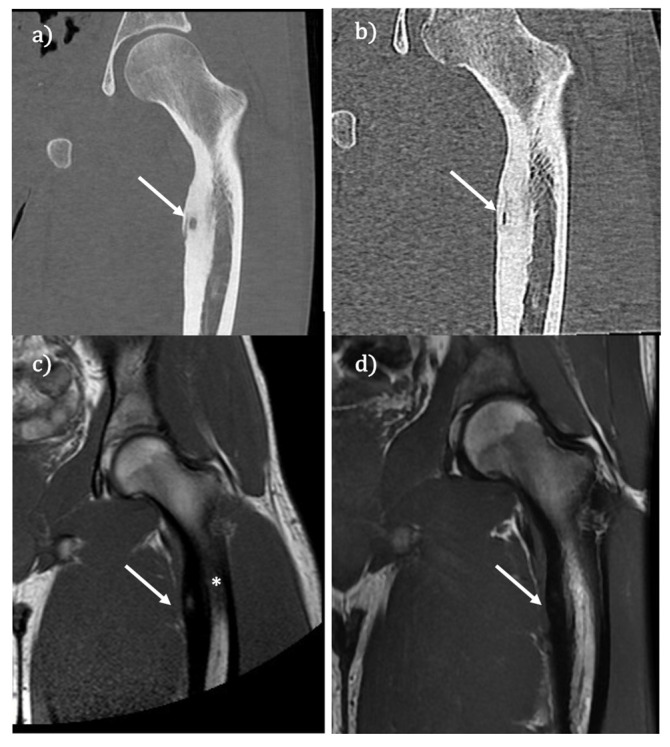
(**a**) Osteoid osteoma (white arrow indicates the cortical reaction and the nidus: TC scans reformatted). (**b**) Follow-up after treatment (1 year): the arrow indicates the vanishing of the nidus. (**c**,**d**) MR T1 weighted scans of the same patients performed before and after treatment. In (**c**), the white arrow indicates the nidus, and the asterisk indicates the bone edema within the bone marrow (bone edema appears hypointense because T1w sequence). (**d**) Disappearance of the nidus and the bone edema (white arrow indicates thee previous bone marrow and nidus).

**Figure 2 jcm-11-01987-f002:**
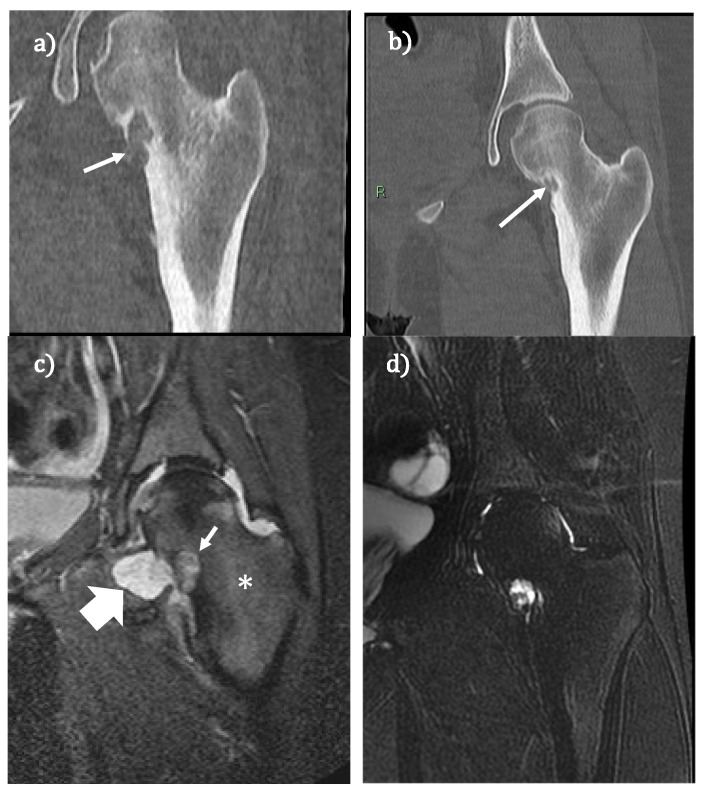
(**a**,**b**) CT scans performed before and after (1 year) RFA osteoblastoma treatment. In (**a**), the white arrow indicates the lesion. In (**b**), the arrow indicates the induced sclerosis of the pathological tissue of the lesion. (**c**,**d**) MR T2 weighted scans with suppression of fat signal of the same patients performed before and after treatment (6 months). In (**c**), the thick white arrow indicates the synovial reaction, the asterisk indicates the bone oedema, and the thin arrow is the lesion. In (**d**), note the disappearance of the synovial reaction and the bone edema.

## Data Availability

Not applicable.

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
