# Peer review of "Post-Procedural Follow-Up of the Interventional Radiology’s Management of Osteoid Osteomas and Osteoblastomas"

_jcm, 2022, doi:10.3390/jcm11071987_

Round 1

Reviewer 1 Report

Dear Authors

Thank you so much  for submitting your  manuscript with title : * Post-procedural follow-up of the interventional radiology’s management of Benign Osteocytic Tumors.  It is an interesting manuscript to be considered for publication.

Howeve+r, there are some questions and comments for this manuscript :

  1. The authors wrote very well, so I am able to read and easily to understand the manuscript. However, there are some grammatical errors that should be corrected.
  2. The authors also need to improve manuscript by :
  3. If the want to make an abbreviation, please inform in the previous paragraphs, such IR, (line 52)
  4. OO in line 109
  5. CT scan
  6. How to cite the references regarding the author guideline

  1. The authors make a title Post-procedural follow-up of the interventional radiology’s management of Benign Osteocytic Tumors, by explain more osteoid osteoma and osteoblastoma. Is it necessary to more focus on the both of tumors or change a title?
  2. The authors wrote that CT scan to diagnose for osteoid osteoma, even though explain the reason why choose MRI and CT for evaluation after treatment, how to compare before and after treatment pre and post MRI.
  3. In conclusion, even though that pain is more subjective symptoms, but it is the main symptoms of osteoid osteoma and osteoblastoma.  How the authors correlate between clinical symptoms and imaging post treatment

Thank you again for submitting your valueable your hard work to J Clinical Medicine.

Warm regards

Author Response

Dear reviewer, thank you so much  for reviewing  our manuscript  "Post-procedural follow-up of the interventional radiology’s management of Benign Osteocytic Tumors".

We are glad to answer you point by point to your observations:

  1. grammatical errors have been corrected
  2. abbreviations have been explained
  3. references have been corrected (unfortunately we had a problem with the template)
  4. you are right, title should be corrected: instead of "osteocytic tumors" we should directly refer to osteoblastomas and osteoid osteomas
  5. we didn't expose so well the MRI role for osteoid osteoma diagnosis, corrected
  6. In conclusion, even if  clinical presentation seems to be more suggestive during the follow-up, we evaluated several MRI and CT aspects that suggest the outcome of procedure (both relapse and resolution) for the reasons we exposed in our manuscript: 

    "Clinical response is the primary outcome to assess the effectiveness of the IR treatment. Anyway, finding a valid method to declare the good outcome of these procedures can be useful because of many reasons:

    1. To confirm by imaging the clinical suspect of recurrence of incomplete treatment
    2. To demonstrate, eventually, any kind of complications
    3. To look for possible differential diagnosis in case of partial or incomplete clinical response"

Kind regards 

Reviewer 2 Report

Review Report

  • In this review, the authors aimed to describe the various possibilities of follow-up protocols and the imaging aspects of recurrence or incomplete treatment of benign Osteocytic Tumors.
  • The paper is interesting, well structured, and correctly organized. The authors have clearly worked hard to detail their study, but I have some comments:
  1. Abstract, line 24: “chirurgical en-block resection’’……What is this mean?
  2. Page 3, line 99: “To confirm by imaging the clinical suspect of recurrence of incomplete treatment”……..or please correct.
  3. Figures 1 and 2: What do you mean by TC and RM?...please correct
  4. Several grammatical errors throughout the manuscript are needed to be corrected.

Author Response

Dear reviewer, thank you for revising our manuscript.

we are glad to clarify point by point your comments:

  1. Abstract, line 24: “chirurgical en-block resection": we didn't talk at length about the chirurgical approach because we focused on the IR one to these lesions, but we wanted to refer to the classical orthopedic surgery the patients usually underwent. 
  2. Page 3, line 99: “To confirm by imaging the clinical suspect of recurrence of incomplete treatment": corrected.
  3. Figures 1 and 2: What do you mean by TC and RM?: of course, computed tomography and magnetic resonance. corrected.
  4. we provided to correct our grammatical errors

Kind regards

Reviewer 3 Report

Authors have presented a short but usefull review about the post-procedural follow-up of the interventional radiology’s management for benign but painful osteocytic tumors, i.e., osteoblastomas and osteoid osteomas.

The text is clearly written in general, although they have to explain more clearly the aim of this review in the abstract ant introduction part.

Author Response

Dear reviewer, thanks for your useful comments.

we provided to explain more clearly the aim of this review in the abstract ant introduction part.

Kind regars

Round 2

Reviewer 3 Report

The authors explained more clearly the aim of this review.

This manuscript is a resubmission of an earlier submission. The following is a list of the peer review reports and author responses from that submission.